# Explaining the increase of incidence and mortality from cardiovascular disease in Indonesia: A global burden of disease study analysis (2000–2019)

**Wigaviola Socha Purnamaasri Harmadha**[1], **Farizal Rizky Muharram**[2], **Renato Simoes Gaspar**[3], **Zahras Azimuth**[5], **Hanif Ardiansyah Sulistya**[5], **Fikri Firmansyah**[5], **Chaq El Chaq Zamzam Multazam**[1], **Muhammad Harits**[5], **Rendra Mahardika Putra**[4]*

1 National Hearth and Lung Institution, Imperial College London, London, United Kingdom, 2 Department of Global Health and Social Medicine, Harvard Medical School, Boston, MA, United States of America, 3 Translational Medicine Department, Universidade Estadual de Campinas (UNICAMP), Campinas, Sao Paulo, Brasil, 4 Department of Cardiology and Vascular Medicine, Airlangga University, Surabaya, Indonesia, 5 Faculty of Medicine, Airlangga University, Surabaya, Indonesia

* rendra.mahardhika@fk.unair.ac.id

**Data Availability Statement:** All relevant data are from the Global Burden study (GBD) by the

## Abstract

### Background

In the last two decades, there has been a discernible shift in the distribution of mortality attributed to cardiovascular disease (CVD) between developing and developed nations; in developed nations, the percentage of deaths caused by CVD decreased from 48% in 1990 to 43% in 2010, while in developing nations, they increased from 18% to 25%. In Indonesia, CVD death has increased substantially and remained elevated in the last ten years. Current behavioral and metabolic risk factors, including hyperglycemia, obesity, dyslipidemia, hypertension, and smoking, enhance the risk of CVD mortality, according to several studies.

### Aims

We undertook a study to determine whether the increase in mortality and incidence of CVD can be attributed to changes in the most common metabolic and behavioral risk factors from 2000 to 2019 across 34 Indonesian provinces.

### Materials and methods

Data from 34 province for CVD incidence and mortality and data on changes in metabolic and behavioural risk factors between 2000 and 2019 in Indonesia were obtained from the Global Burden study (GBD) by The Institute of Health Metrics and Evaluation (IHME). A statistical model was applied to calculate the fatalities attributable to the risk factors change using Population attributable fractions (PAF) and baseline year death numbers. Furthermore, we ran multivariate regressions on Summary Exposure Value of risk factors associated with the increasing mortality, incidence rates in a lag year analysis. R software used to measure heteroscedasticity-consistent standard errors with coeftest and coefci. Covariates

Institute of Health Metrics and Evaluation (IHME): https://www.healthdata.org/research-analysis/gbd. Please see the paper and Supporting Information files for details.

**Funding:** The author(s) received no specific funding for this work.

**Competing interests:** The authors have declared that no competing interests exist.

were added to adjusted models, including the Socio-demographic Index, Primary health care facilities coverage, and GDP per capita.

## Results

The age-standardized mortality rate for CVD from 2000 to 2019 in Indonesia, increased from 356.05 to 412.46 deaths per 100,000 population among men and decreased from 357.52 to 354.07 deaths per 100,000 population among women, resulting in an increase of 270.928 per 100,0000 inhabitants of CVD deaths. In the same period, there was an increase in exposure to risk factors such as obesity by +9%, smoking by +1%, dyslipidemia by +1.3%, hyperglycemia by +2%, and hypertension by +1.2%. During this time span, an additional 14,517 men and 17,917 women died from CVD, which was attributable to higher obesity exposure. We apply multivariate regression with province-fixed and year-fixed analysis and find strong correlation between hyperglycemia in women (6; 95%CI 0 to 12, death per 1-point increase in hyperglycemia exposure) with an increasing death rate in ischemic heart disease. We also performed a year lag analysis and discovered a robust association between high low density lipoprotein (LDL) levels in men and women and the growing incidence of ischemic heart disease. The association between a 10-year lag of high LDL and the incidence of ischemic heart disease was five times stronger than that observed for other risk factors, particularly in men (5; 95%CI 2 to 8, incidence per 1-point increase in high LDL exposure).

## Conclusion

Hyperglycemia in women is an important risk factor associated with increasing mortality due to Ischemic Heart Disease (IHD) in Indonesia This study also revealed that the presence of high LDL in both men and women were associated with an increase incidence of IHD that manifested several years subsequent to exposure to the risk factor. Additionally, the highest cardiovascular death portion were attributed to obesity. These findings suggest that policymakers should control high LDL and hyperglycemia 10 years earlier prior to the occurrence of IHD and employ personalized therapy to regulate associated risks.

## Introduction

Cardiovascular diseases (CVD) are the main cause of mortality worldwide with a burden greater than any other disease. In 2010, cardiovascular diseases accounted for 12% of 2482 million disability-adjusted life years (DALYs) and 30% of 52 million deaths worldwide. In the recent two decades, the cardiovascular disease burden has transitioned from developed to developing countries, particularly in low- and middle-income nations (LMICs) such as Indonesia. In developed countries, the proportion of fatalities attributable to cardiovascular disease declined from 48% in 1990 to 43% in 2010, whereas it climbed from 18% to 25% in developing countries during the same time period [1]. The burden of CVD in developing countries is starting to increase, while in well-developed countries, it is the opposite. This change indicates we must investigate what transformation could contribute to those changes.

In the present age, the prevalence of frequent behavioral and metabolic risk factors, such as hyperglycemia, obesity, dyslipidemia, hypertension, and smoking, have been acknowledged by

several studies to highly increase the risk of developing cardiovascular disease. Previous study in Brazil found that the mortality rate experienced decreased from 2005 to 2017, and further analysis has shown that hyperglycemia was the only risk factor that had a significant and reliable correlation with CVD mortality, especially in women [2]. Another study conducted in the United States found that the mortality rate fell by more than 40% between 1980 and 2000 as a result of changes in risk factors, which led to approximately 149,635 fewer deaths from coronary heart disease [3]. Prior research has found that among these risk factors hypertension was the most common risk factor leading to a larger number of cases due to cardiovascular events in low-income nations (LICs) [4]. Meanwhile, most cardiovascular disease deaths can be attributed to a small number of common modifiable risk factors. Thus, health policies should focus on risk factors that have the most impact on a global scale to prevent cardiovascular disease and death, with additional emphasis on risk factors that are most influential on each country [5].

According to the results of the survey from the Indonesian National Basic Health Research, the prevalence of smoking, a high-risk food diet, and a low HDL cholesterol level was higher among men, while a low physical activity level, the presence of mental-emotional disorders, obesity, a high waist circumference (WC), a high waist-to-hip ratio (WtHR), hypertension, diabetes, a high total cholesterol level, and a high LDL cholesterol level were more prevalent among women. The prevalence of individuals with at least four risk factors was approximately 22% among men and 18% among women. Notably, these proportions were found to be greater in individuals who had previously been diagnosed with cardiovascular diseases [6].

Although the present study indicates that there have been increases in the use of prescription medications to treat hypertension, diabetes mellitus, and dyslipidemia in both women and men, the management was suboptimal—a considerable number of proportions are not treated accordingly to current guidelines and do not receive adequate control [7]. There is a sex gap and age disparities in terms of the metabolic factors. Previous study done in the United States showed that men were less likely to have hypertension and diabetes under control, whereas women were less likely than males to have dyslipidemia under control. Nonetheless, women were less likely to receive adequate treatment for these disorders. Overall, there are gender inequalities in the prevalence of risk factors and CVD mortality [8].

Using province-level data from 2000 to 2019 in Indonesia, we examined the association between various prevalent behavioral and metabolic risk factors (hypertension, hyperglycemia, obesity, dyslipidemia, obesity, and smoking) and CVD. Furthermore, we assessed the proportion of the escalation of CVD mortality between 2000 and 2019 that might be attributed to changes in the prevalence of such risk factors at the province level. Overall, our findings may assist policymakers in implementing more effective methods to address these risk factors, thereby reducing the prevalence of cardiovascular disease in Indonesia.

## Methods

### Study model and data sources

In this retrospective investigation, a multilevel analysis was conducted in stages. In the initial phase, we obtain data for the dependent variable from the CVD outcome, including data on cardiovascular diseases, ischemic heart disease, and ischemic stroke mortality rate and incidence rate per 100,000. By accumulating data on Summary exposure value (SEV) for the high BMI, hyperglycemia, high systolic blood pressure, dyslipidemia, and smoking, independent variable data were collected. The Institute of Health Metrics and Evaluation (IHME), an impartial health research center at the University of Washington, collected all the dependent and independent variables from 34 provinces in Indonesia between 2000 and 2019 using

publicly accessible datasets. Next, covariates on the GINI index, health facilities, and GDP per capita were added. Covariates were included in the multivariate regression analysis.

## Metabolic and behavioural risk factors

Exposure to risk factors contributing to the escalation number of Disability-Adjusted Life Years (DALYs) and mortality rate attributed to CVD namely hypertension, dyslipidemia, hyperglycemia, obesity, smoking was computed as the Summary Exposure Value (SEV). A measure of a population's risk factor exposure that considers the extent of exposure by risk level as well as the severity of that risk's contribution to disease burden. SEV has a value of zero when there is no extra risk in a population and a value of one when the population is at the maximum degree of risk; we present SEV on a scale of 0% to 100% to indicate that it is risk-weighted prevalence. The parameter data for the dependent variables (hyperglycemia, obesity, dyslipidemia, hypertension, and smoking) were obtained through the examination of high glucose level, high body mass index, high low-density lipoprotein, high systolic blood pressure and smoking from 34 provinces in Indonesia. High blood pressure when SBP is greater than 110–115 mm Hg; high body mass index when BMI is greater than 21.0–24.0 kg/m2; high fasting plasma glucose when greater than 4.9–5.3 mmol/L; dyslipidemia when total cholesterol is greater than 3.8–4.0 mmol/L; and smoking any tobacco product.

## Cardiovascular diseases outcome; incidence and death

Mortality rates, the number of deaths per 100,000 people, and incidence rate owing to cardiovascular diseases, ischemic heart disease, or stroke by gender, province, and year, were used as dependent variables throughout the study. Age-standardized data were used to compare populations with different age structures. Population characteristics were statistically transformed to match those of a reference population. Year lag analysis was added for incidence to determine the rate at which new events occur in a defined population owing to the risk factors. From 2000 to 2019, data were collected continuously.

## Covariates

In this study, covariates were used to examine the analysis affecting the independent variables. First, we compiled the Socio-demographic Index from 2000 to 2019 for each province in Indonesia, totaling 34 provinces. The values vary from 0 to 1, with 0 being the greatest disparity and 1 indicating equality. Data were acquired from the Indonesian Ministry of Health profile, a publicly open-source data. GDP Per Capita, is the next covariate to be considered. Finally, we included the number of health centers from the Indonesian Ministry of Health profile.

## Death attributed to risk factors between 2000 to 2019

We calculated how much of the increase in cardiovascular disease death between 2000 and 2019 could be attributed to the prevalence of behavioral and metabolic risk factors during the same time. IHME provided the sex-specific population attributable fractions (PAF) of cardiovascular disease (CVD) events attributed to each risk factor (hyperglycemia, obesity, dyslipidemia, high systolic blood pressure, and tobacco). Prior research [3, 9] has produced an algorithm for calculating the number of fatalities averted or delayed due to changes in risk variables. The equation also includes the population attributable fractions (PAF) where it is used to determine the effect of changes in the prevalence of hyperglycemia, obesity, dyslipidemia, hypertension, and smoking to CVD. To compute this, the number of fatalities from each category of CVD in the basis year of 2005 was multiplied by the difference between the PAF from

risk factors in the base year (2000) and the final year (2019). This figure was then multiplied by -1 to determine the number of fatalities that occurred because of changes in risk variables, as opposed to the number of deaths that were averted or postponed. Hence, a positive number represents the number of fatalities caused by a certain risk factor, and a negative number represents the number of deaths that were prevented/delayed. For instance, the PAF of High Body Mass Index increased from 0.1 to 0.18 and the baseline number of death due to CVD in 2000 were 380553, then, the additional cardiovascular disease deaths in 2019 that were related to an increasing prevalence of high BMI were consequently computed as follows:

$$\text{Number of deaths from CVD attributed due to changes of risk factors}$$
$$= 380553 \text{ x } (0.1 - 0.18) = -30444 \text{ x} -1$$
$$= 30444$$

## Statistical analysis

A multivariate regression model using R software evaluated CVD outcome data as the dependent variable, SEV of each risk factor as the main independent variable, and the adjusted covariate. This study included CVD mortality and incidence rates per 100,000 people. Pearson clustered standard error tests avoided were used to autocorrelation. State fixed effects accounted for unobserved cultural, geographic, and historical characteristics between states but fixed over time. Year fixed effects controlled unseen elements that vary over time but are comparable across states. Fixed-effect models analyze aggregated health data well. Fixed effects control province variation that may affect study outcomes. Year fixed effects account for unobserved factors that fluctuate over time but affect all provinces equally. This controls time-specific effects on outcomes. Fixed-effects models handle unobserved variables that may confound aggregated health data, making the study more robust. This method isolates and estimates the variables of interest, improving the risk factor-health outcome link.

## Ethic statement

This research did not require permission from an Ethics committee because it utilized solely data from publically accessible secondary databases.

## Patient and public involvement statement

It was neither suitable nor feasible to include patients or the general public in the research's conception, implementation, reporting, or dissemination plans.

## Results

The mortality rate caused by cardiovascular disease (CVD) across 34 provinces in Indonesia continuously increased and peaked between 2000 to 2019; Age-standardized DALYs and Mortality rate for cardiovascular diseases per 100.000 population increased. DALYs increased from 7,727 to 7,777 (+50.3%), and the mortality rate increased from 357.84 to 383.26 (+25.4%), respectively (Fig 1). Mortality rate for cardiovascular disease increased from 356.05 to 412.46 deaths per 100,000 population among men and decreased from 357.52 to 354.07 deaths per 100,000 population among women, resulting in an increase of 270.928 per 100,0000 inhabitants of cardiovascular disease deaths. The highest mortality in CVD can be seen in 2015; nevertheless, we have seen a drop in CVD mortality since 2015. From 2015 to 2019, the CVD mortality rate fell from 394.38 per 100,000 population to 383.26 per 100,000 population (Fig 1). Over the past few years, stroke and ischaemic heart disease have been Indonesia's most dominating cardiovascular diseases.

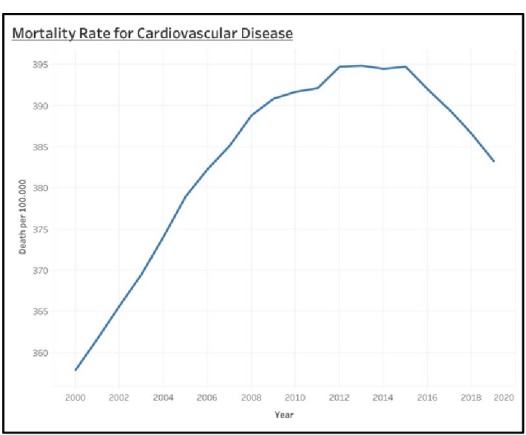 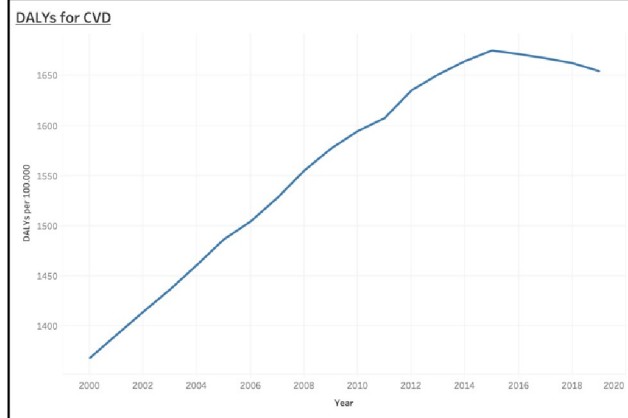

**Fig 1. Age-standardized mortality rate and DALYs of cardiovascular diseases.** The Global Health Data Exchange included age-standardized country data for both sexes (GHDx).

Between 2000 and 2019, the SEV rates for all risk factors climbed (Fig 2). Regarding the SEV rate from 2000 to 2019 in Indonesia, it is shown that Hypertension was the most dominated risk factor compared to the others, followed by dyslipidemia, with a stagnant line graph. Hypertension SEV rate was stabilized from 37,8 to 39 (+1,2%), dyslipidemia and smoking risk factor slowly increasing from 22,2 to 23,6 (+1,4%) and 11,4 to 12,4 (+1%), respectively, over the past 19 years. The SEV rate of obesity is shown to be the least risk factor rate compared to other risk factors at the beginning of 2000, whereas in 2019, the SEV rate of obesity outnumbered both high fasting plasma glucose and smoking, then remained escalated in the end of the year of 2019. Obesity's SEV rate dramatically soared up from 6,7 to 16 (+9,3%) and high fasting plasma glucose grew slowly from 6,9 to 8,9 (+7%).

Generally speaking, there was an increase in mortality that is attributed from behavioral and metabolic risk factors over the past 19 years, namely hypertension, high fasting plasma glucose, dyslipidemia, obesity, and current smoking. Hypertension had been dominating the most attributable death for cardiovascular diseases among other risk factors for the past 19 years although the trend seems stagnant. In 2000, the proportion of death attributed from hypertension was 67% or accounted for 255,845, while in 2019, it increased to be 69% which is accounted for 449,376. Nevertheless, there is a notable recognition in high body mass index or obesity which is showed to dramatically increasing from 9% to 18% between 2000 to 2019 which is accounted for 37,864 and 118,188. In addition, high fasting plasma glucose is also depicted to be increase with the similar trend (Fig 3).

In overview, each risk factor's SEV differs between men and women. The female group dominated obesity, dyslipidemia, and hypertension, while the male group dominated tobacco usage and hyperglycemia. Hypertension has a considerable gender gap as the risk factor with the highest exposure, indicating that women are more susceptible to hypertension than males. Yet, between 1990 to 2019, the proportion of males with high blood pressure grew. Between 1990 and 2019, the gender disparity in SEV rate decreased from 13.36 to 1.72, while hyperglycemia remained stable at 8.47 and 9.53. The SEV rate disparity in tobacco use between men and women was 31% in 1990 and has increased by almost 2% to 33.67% in 2019.

Table 1 provides an overview of various risk factors and their associated impact on cardiovascular diseases, specifically ischemic heart diseases and ischemic stroke. Each risk factor is evaluated based on its Population Attributable Fraction (PAF), representing the proportion of diseases cases in the population that can be attributed to that specific risk factor. The PAF values are presented as ranges, indicating the best estimates with minimum and maximum values.

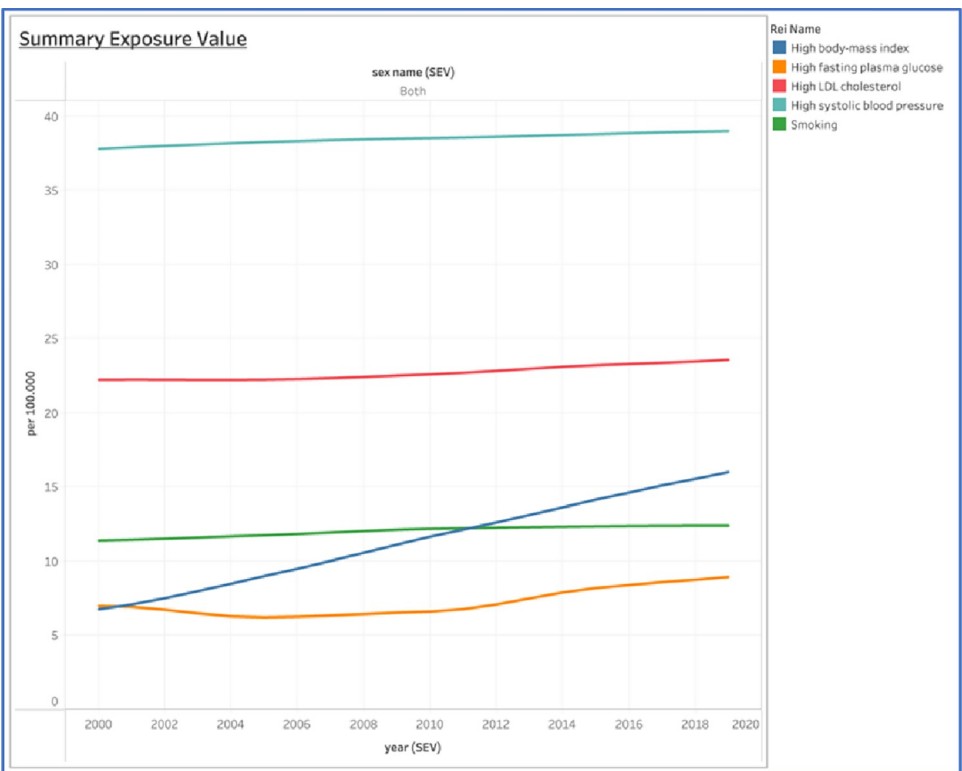

**Fig 2. Summary Exposure Value (SEV) of risk factors in Indonesia between 2000 to 2019.** The Global Health Data Exchange included age-standardized country data for both sexes (GHDx).

We calculated the number of deaths that occured due to changes in the PAF of risk factors (Table 1) using the equation that previously made from study done in America [7]. Obesity was the risk factor that accounted for the highest number of deaths due to Cardiovascular, with a predicted number of death 14517 men and 17917 women. Looking specifically to death attributed from obesity, it shown that around 35% of additional death in cardiovascular disease were due to Ischaemic Heart Disease (IHD). We discovered a significant gender disparity in SEV smoking, with an additional 3629 deaths in men and only 398 deaths in women.

The correlation between multiple risk factors and the mortality rate of cardiovascular disease, specifically ischemic heart disease and ischemic stroke, was examined using multivariate regression analyses (Table 2). The estimated coefficients and their associated 95% confidence intervals (2.5% and 97.5% bounds). This table reveals that high glucose is substantially associated with the mortality rate of women with ischemic heart disease (8; 95%CI 2 to 13, death per 1-point increase in hyperglycemia exposure). In contrast, dyslipidemia is significantly associated with mortality in males (2; 95%CI 0 to 3, death per 1-point increase in hyperglycemia exposure). On the other hand, males with ischemic stroke have a significant association with elevated systolic blood pressure (1; 95%CI 0 to 2, death per 1-point increase in hyperglycemia exposure). Despite the reported associations between high blood sugar in women and high LDL in men with ischemic heart disease, we found no association between mortality CVD and risk factors.

We expanded our analysis and included covariates such as the Socio-demographic Index, primary health care facility coverage, and GDP per capita from each province; the relationship between multiple risk factors and the mortality rate of cardiovascular disease, specifically

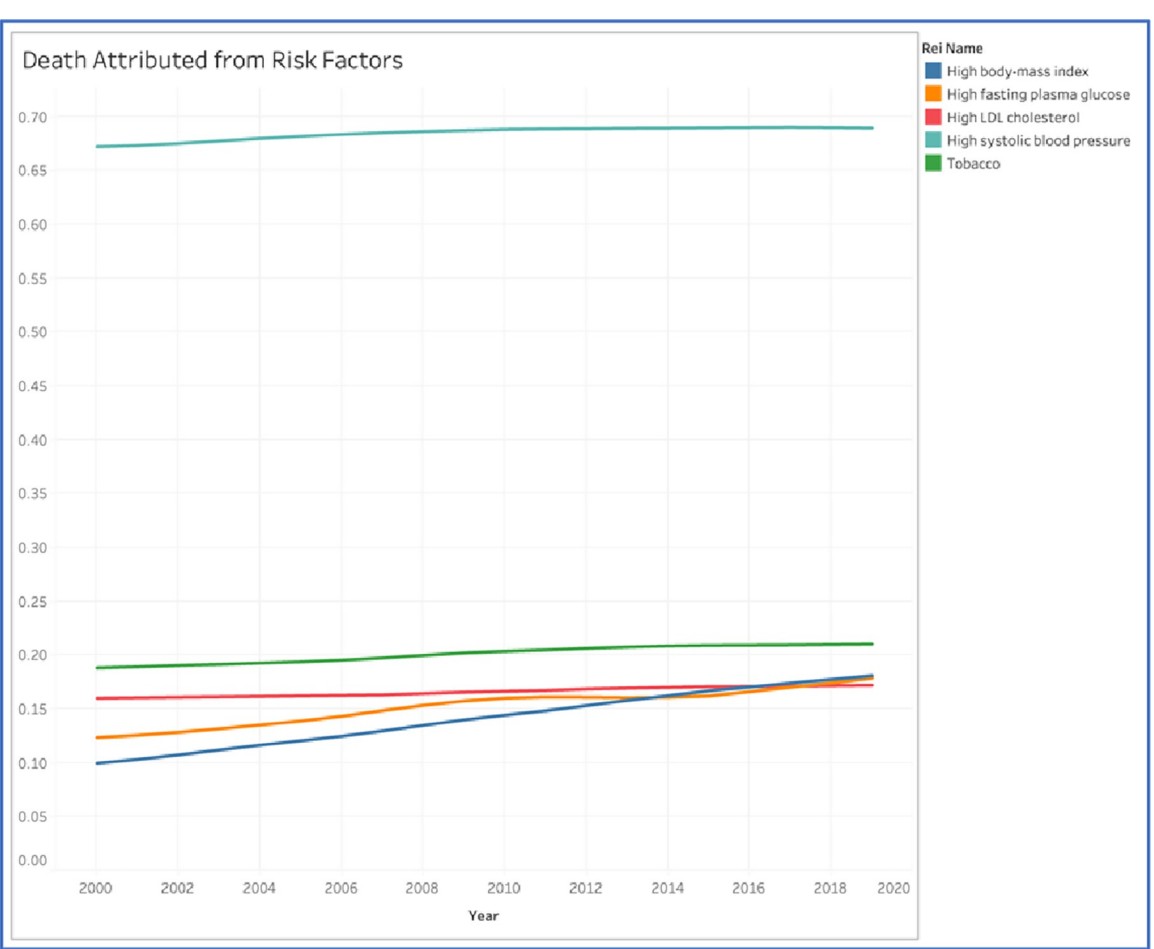

**Fig 3. Death attributed in cardiovascular diseases by risk factors in Indonesia between 2000 to 2019 in percentage.** The Global Health Data Exchange included age-standardized country data for both sexes (GHDx).

ischemic heart disease and ischemic stroke, was investigated using multivariate regression analyses (Table 3). According to this table, elevated hyperglycemia is still related with a higher mortality rate in women with ischemic heart disease (6; 95%CI 0 to 12, death per 1-point increase in hyperglycemia exposure), however, high LDL does not related with IHD mortality in men. Males with ischemic stroke, on the other hand, had a significant connection with raised systolic blood pressure (1; 95%CI 0 to 2). There is a negative correlation in high BMI and Ischemic Stroke that might be an artefact that caused due to autocorrelation.

The tables below (Tables 4 and 5) depict a lag time study of risk variables for the incidence of ischemic heart disease from 2000 to 2019. The findings of this research, which used robust standard error regression, demonstrate that there is a significant association between exposure to high LDL risk factors and ischemic heart disease incidence in both men and women several years prior. Other risk variables did not show this. In men with high LDL, we identified a significant connection in 2 years prior (4.59; 95% CI 1.98 to 7.2), 5 years prior (4.68; 95% CI 199 to 7.37), 8 years prior (4.99; 95% CI 2.23 to 7.75), and 10 years prior (4.99; 95% CI 2.23 to 7.75). In women, we identified a significant connection in 2 years prior (1.39; 95% CI 0.27 to 2.51). 5 years prior (1.4; 95% CI 0.2 to 2.6), 8 years prior (1.53; 95% CI 0.21 to 2.84), 10 years prior (1.56; 95% CI 0.22 to 2.91).

**Table 1. Death from cardiovascular diseases, Ischaemic heart diseases and Ischaemic stroke that were attributed as a result of changes in risk factors in Indonesia, 2000 to 2019.**

| Risk Factor | Population Attributable Fraction (PAF) | | | | | | Death Attributed to Changes in Risk Factor | | |
|---|---|---|---|---|---|---|---|---|---|
| | Best (minimum to maximal) estimates | | | | | | Best (minimum to maximal) estimates | | |
| | Cardiovascular Disease | | Ischaemic Heart Disease | | Ischaemic Stroke | | Cardiovascular Diseases | Ischaemic Heath Disease | Ischaemic Stroke |
| | 2000 | 2019 | 2000 | 2019 | 2000 | 2019 | | | |
| SEV High BMI | | | | | | | | | |
| Men | 0.09 (0.05–0.18) | 0.17 (0.1–0.25) | 0.08 (0.03–0.15) | 0.15 (0.08–0.23) | 0.045 (0.01–0.09) | 0.09 (0.05–0.15) | **14517** | **5125** | **1326** |
| Women | 0.1 (0.5–0.18) | 0.19 (0.12–0.27) | 0.08 (0.04–0.15) | 0.16 (0.1–0.24) | 0.048 (0.02–0.09) | 0.094 (0.05–0.14) | **17917** | **5150** | **1648** |
| Both | 0.1 (0.04–0.18) | 0.18 (0.11–0.25) | 0.08 (0.03–0.15) | 0.16 (0.1–0.2) | 0.05 (0.02–0.09) | 0.092 (0.05–0.14) | **30444** | **11008** | **2743** |
| SEV High Glucose | | | | | | | | | |
| Men | 0.13 (0.09–0.18) | 0.19 (0.13–0.27) | 0.15 (0.09–0.24) | 0.21 (0.12–0.34) | 0.17 (0.08–0.35) | 0.24 (0.11–0.49) | 10888 | 4393 | 2062 |
| Women | 0.11 (0.08–0.16) | 0.17 (0.11–0.24) | 0.14 (0.08–0.21) | 0.2 (0.11–0.34) | 0.14 (0.07–0.29) | 0.2 (0.09–0.43) | 11944 | 3863 | 2150 |
| Both | 0.12 (0.09–0.17) | 0.18 (0.12–0.25) | 0.14 (0.09–0.22) | 0.2 (0.12–0.33) | 0.15 (0.07–0.31) | 0.21 (0.1–0.47) | 22833 | 8256 | 3918 |
| SEV High LDL | | | | | | | | | |
| Men | 0.18 (0.14–0.22) | 0.19 (0.14–0.23) | 0.3852 (0.385–0.381) | 0.3857 (0.31–0.47) | 0.143 (0.07–0.28) | 0.145 (0.07–0.28) | 1814 | 36 | 58 |
| Women | 0.14 (0.11–0.18) | 0.15 (0.11–0.2) | 0.363 (0.28–0.45) | 0.364 (0.0.28–0.45) | 0.142 (0.06–0.29) | 0.144 (0.06–0.31) | 1990 | 64 | 71 |
| Both | 0.16 (0.13–0.2) | 0.17 (0.13–0.21) | 0.37 (0.30–0.46) | 0.38 (0.3–0.46) | 0.142 (0.06–0.29) | 0.144 (0.06–0.29) | 3805 | 275 | 130 |
| SEV High SBP | | | | | | | | | |
| Men | 0.64 (58–69) | 0.67 (0.61–0.71) | 0.61 (0.54–0.69) | 0.64 (0.57–0.71) | 0.58 (0.46–0.68) | 0.6 (0.49–0.70) | 5444 | 2196 | 589 |
| Women | 0.70 (64–74) | 0.71 (0.66–0.76) | 0.68 (0.59–0.75) | 0.7 (0.6–0.78) | 0.64 (0.52–0.74) | 0.65 (0.52–0.77) | 1990 | 1287 | 358 |
| Both | 0.67 (0.62–0.72) | 0.69 (0.63–0.73) | 0.65 (0.56–0.72) | 0.67 (0.58–0.74) | 0.61 (0.49–0.72) | 0.63 (0.51–0.73) | 7611 | 2752 | 1306 |
| SEV Smoking | | | | | | | | | |
| Men | 0.28 (26–30) | 0.30 (0.28–0.32) | 0.36 (0.34–0.38) | 0.4 (0.38–0.42) | 0.23 (0.21–0.25) | 0.27 (0.24–0.29) | 3629 | 2928 | 1178 |
| Women | 0.032 (0.02–0.04) | 0.034 (0.03–0.4) | 0.04 (0.04–0.06) | 0.13 (0.11–0.15) | 0.03 (0.02–0.03) | 0.07 (0.05–0.83) | 398 | 5794 | 1433 |
| Both | 0.15 (0.14–0.17) | 0.17 (0.15–0.2) | 0.21 (0.19–0.23) | 0.29 (0.26–0.31) | 0.12 (0.10–0.14) | 0.16 (0.14–0.2) | 7611 | 11008 | 2612 |
| Number of Death in Baseline Year (2000) | | | | | | | | | |
| Men | | | | | | | 181472.366 | 73223.0261 | 29470.2083 |
| Women | | | | | | | 199081.286 | 64383.9774 | 35842.128 |
| Both | | | | | | | 380553.652 | 137607.003 | 65312.3363 |

[a] Calculated by the Institute for Health Metrics and Evaluation. Data are shown for the best estimates as well as the minimum and maximum values of PAF.

[b] Calculated as described in Materials and Methods.

SEV: Summary Exporsure Value. BMI: Body Mass Index. LDL: Low-density lipoprotein. SBP: Systolic Blood Pressure.

**Table 2. Associations between SEV for risk factors and mortality rates for CVD, Ischaemic heart disease, and stroke for men and women in Indonesia, unadjusted models.**

| | Men | | | | | | Women | | | | | |
| --- | --- | --- | --- | --- | --- | --- | --- | --- | --- | --- | --- | --- |
| | Cardiovascular Diseases (Coefficient, 95% CI) | p | Ischaemic Heart Disease (Coefficient, 95%CI) | p | Ischaemic Stroke (Coefficient, 95%CI) | p | Cardiovascular Diseases (Coefficient, 95% CI) | p | Ischaemic Heart Disease (Coefficient, 95%CI) | p | Ischaemic Stroke (Coefficient, 95%CI) | p |
| SEV High Plasma Glucose | 0.03 (-15.11 to 15.18) | 0.99 | -4.46 (-14.55 to 5.62) | 0.38 | 1.72 (-3.21 to 5.75) | 0.57 | 15.37 (-1.39 to 32.14) | 0.07 | **7.86 (2.37 to 13.35) \*** | **0.05** | 0.64 (-3.85 to 5.13) | 0.77 |
| SEV high BMI | -2.73 (-15.11 to 15.18) | 0.32 | -0.6 (-2.53 to 1.32) | 0.53 | -0.68 (-2.06 to 0.68) | 0.54 | -4.29 (-10.8 to 2.21) | 0.19 | -0.82 (-2.04 to 0.38) | 0.18 | -0.37 (-2.38 to 1.64) | 0.71 |
| SEV high LDL | 1.86 (-2.19 to 5.922) | 0.36 | **2.25 (0.91 to 3.58) \*\*\*** | **0.0009** | 0.61 (-0.28 to 1.52) | 0.18 | -1.24 (-5.77 to 3.28) | 0.58 | 0.05 (-1.61 to 1.72) | 0.94 | 0.79 (-0.47 to 2.06) | 0.22 |
| SEV high SBP | 2.46 (-2.63 to 7.56) | 0.34 | 0.67 (-1.42 to 2.21) | 0.67 | **1.15 (0.01 to 2.29) \*** | **0.04** | 1.13 (-3.68 to 5.95) | 0.64 | 1.45 (-0.97 to 3.89) | 0.24 | 0.01 (-2.08 to 2.11) | 0.98 |
| SEV smoking | -1.25 (-8.85 to 6.34) | 0.74 | -0.1 (-3.42 to 3.21) | 0.95 | -0.01 (-1.77 to 1.73) | 0.98 | 29.43 (-29.62 to 88.5) | 0.32 | 3.71 (-16.83 to 24.25) | 0.72 | 7.14 (-10.91 to 25.19) | 0.43 |

Mortality and SEV were per 100,000 individuals. Estimates and 95% confidence intervals (CI) are provided in the figure

\* p<0.05

\*\* p<0.01

\*\*\* p<0.001

SEV: summary exposure value. LDL: Low density lipoprotein. BMI: body mass index. SBP: systolic blood pressure.

**Table 3. Associations between SEV for risk factors and mortality rates for CVD, Ischaemic heart disease, and stroke for men and women in Indonesia, adjusted models.**

| | Men | | | | | | Women | | | | | |
| --- | --- | --- | --- | --- | --- | --- | --- | --- | --- | --- | --- | --- |
| | Cardiovascular Diseases (Coefficient, 95% CI) | p | Ischaemic Heart Disease (Coefficient, 95%CI) | p | Ischaemic Stroke (Coefficient, 95%CI) | p | Cardiovascular Diseases (Coefficient, 95% CI) | p | Ischaemic Heart Disease (Coefficient, 95%CI) | p | Ischaemic Stroke (Coefficient, 95%CI) | p |
| SEV High Plasma Glucose | 6.07 (-6.01 to 18.1) | 0.32 | -5.27 (-17.6 to 7.14) | 0.4 | 2.29 (-1.35 to 5.93) | 0.21 | 12.8 (-0.14 to 25.8) | **0.05\*** | **6.01 (-0.11 to 12.14)** | **0.005\*** | 0.12 (-4.5 to 4.74) | 0.95 |
| SEV high BMI | -5.86 (-15.2 to 3.49) | 0.21 | -0.37 (-6.38 to 5.63) | 0.9 | **-2.32 (-4.68 to 0.037)** | **0.05\*** | 4.38 (-1.58 to 10.35) | 0.14 | 0.43 (-1.63 to 2.5) | 0.67 | 0.86 (-0.94 to 2.68) | 0.34 |
| SEV high LDL | -2.29 (-5.51 to 0.92) | 0.16 | 1.76 (-0.87 to 4.4) | 0.18 | -0.65 (-1.46 to 0.16) | 0.11 | 3.69 (-1.28 to 8.67) | 0.14 | 0.91 (-1.73 to 3.56) | 0.49 | 1.12 (-0.24 to 2.48) | 0.1 |
| SEV high SBP | 3.8 (-0.16 to 7.78) | 0.06 | 0.46 (-1.62 to 2.55) | 0.66 | **1.4 (0.42 to 2.39)** | **0.004\*\*** | 1.98 (-1.47 to 5.45) | 0.26 | 1.57 (-0.9 to 4.05) | 0.21 | 0.35 (-1.51 to 2.22) | 0.7 |
| SEV smoking | -0.78 (-6.64 to 5.07) | 0.79 | -0.39 (-3.44 to 2.66) | 0.8 | 0.26 (-1.05 to 1.59) | 0.69 | 20.2 (-29.8 to 70.45) | 0.42 | 7.25 (-14.39 to 28.9) | 0.51 | 5.62 (-10.4 to 21.69) | 0.4 |

Variables are included in the model: Socio-demographic Index, primary health care facilities, GBD per capita, state and year fixed effects and SEV for risk factors.

Mortality and SEV were per 100,000 individuals. Estimates and 95% confidence intervals (CI) are provided in the figure

\* p<0.05

\*\* p<0.01

\*\*\* p<0.001

SEV: summary exposure value. LDL: Low density lipoprotein. BMI: body mass index. SBP: systolic blood pressure.

**Table 4. Lag-time analysis between the risk factors and the incidence of ischaemic heart diseases in men in Indonesia from 2000 to 2019.**

| | Men | | | | | | | |
|---|---|---|---|---|---|---|---|---|
| | 2-year lag (Coefficient, 95%CI) | p | 5-year lag (Coefficient, 95%CI) | p | 8-year lag (Coefficient, 95%CI) | p | 10-year lag (Coefficient, 95%CI) | p |
| SEV High Plasma Glucose | 1.1 (-6.01 to 8.22) | 0.76 | 0.97 (-6.68 to 8.62) | 0.8 | -1.17 (-9.34 to 6.99) | 0.77 | -1.74 (-9.99 to 6.5) | 0.67 |
| SEV high BMI | -0.57 (-2.43 to 1.27) | 0.54 | -0.62 (-2.68 to 1.43) | 0.55 | -0.74 (-3.1 to 1.61) | 0.53 | -0.82 (-3.39 to 1.75) | 0.53 |
| SEV high LDL | 4.59 (1.98 to 7.2) | 0.0005*** | 4.68 (1.99 to 7.37) | 0.0006*** | 4.99 (2.23 to 7.75) | 0.0004*** | 5.10 (2.35 to 7.85) | 0.0003*** |
| SEV high SBP | -2.03 (-5.55 to 1.49) | 0.25 | -2.05 (-5.94 to 1.73) | 0.26 | -2.06 (-5.87 to 1.73) | 0.28 | -2.1 (-5.94 to 1.73) | 0.28 |
| SEV smoking | -4.6 (-10.9 to 1.63) | 0.14 | -4.6 (-11.4 to 2.13) | 0.16 | -4.62 (-11.38 to 2.13) | 0.17 | -4.66 (-11.4 to 2.13) | 0.17 |

Estimates and 95% confidence intervals (CI) are provided in the figure

* $p < 0.05$

** $p < 0.01$

*** $p < 0.001$

SEV: summary exposure value. LDL: Low density lipoprotein. BMI: body mass index. SBP: systolic blood pressure.

## Discussion

### CVD burden trend in developed and developing countries

Cardiovascular disease (CVD) is still the leading cause of mortality worldwide. A greater number of CVD deaths is found in countries going through economic transition such as India, China, and Russia. The explanation for this uptrend in CVD burden may be caused due to economic prosperity, which is associated with a high-fat diet, obesity, diabetes mellitus, tobacco use, and a sedentary lifestyle, all well-defined risk factors for developing CV disease [10]. This trend follows the same pattern observed in developing countries like Indonesia, where there has been a substantial incline in CVD incidence, prevalence, and mortality rates (RISKES-DAS). Previous research in Indonesia indicated that approximately 470,000 people die annually due to CVD [11]. These findings also align with the trend observed in other developing countries such as China, where the mortality has been dramatically increased since the 1980s [12].

**Table 5. Lag-time analysis between the risk factors and the incidence of ischaemic heart diseases in women in Indonesia from 2000 to 2019.**

| | Women | | | | | | | |
|---|---|---|---|---|---|---|---|---|
| | 2-year lag (Coefficient, 95%CI) | p | 5-year lag (Coefficient, 95%CI) | p | 8-year lag (Coefficient, 95%CI) | p | 10-year lag (Coefficient, 95%CI) | p |
| SEV High Plasma Glucose | -0.19 (-1.99 to 1.6) | 0.83 | 0.28 (-1.87 to 2.44) | 0.79 | -0.98 (-3.07 to 1.11) | 0.35 | -1.16 (-3.27 to 0.94) | 0.27 |
| SEV high BMI | -0.34 (-1.06 to 0.38) | 0.35 | -0.36 (-1.14 to 0.41) | 0.35 | -0.29 (-1.1 to 0.52) | 0.47 | -0.32 (-1.19 to 0.53) | 0.45 |
| SEV high LDL | 1.39 (0.27 to 2.51) | 0.01* | 1.4 (0.2 to 2.6) | 0.02** | 1.53 (0.21 to 2.84) | 0.02* | 1.56 (0.22 to 2.91) | 0.02* |
| SEV high SBP | -0.04 (-1.32 to 1.22) | 0.93 | -0.1 (-1.4 to 1.19) | 0.87 | -0.16 (-1.54 to 1.2) | 0.81 | -0.18 (-1.58 to 1.21) | 0.79 |
| SEV smoking | -6 (-15.7 to 3.6) | 0.21 | -6.69 (-16.69 to 3.29) | 0.18 | -6.76 (-17.12 to 3.59) | 0.2 | -6.77 (-17.43 to 3.87) | 0.21 |

Estimates and 95% confidence intervals (CI) are provided in the figure

* $p < 0.05$

** $p < 0.01$

*** $p < 0.001$

SEV: summary exposure value. LDL: Low density lipoprotein. BMI: body mass index. SBP: systolic blood pressure.

However, this pattern contrasts with well developed nations like the United States and Brazil, where there has been a decline in fatalities related to cardiovascular disease [2, 9]. Research conducted in Brazil revealed a decrease in cardiovascular disease mortality between 2005 and 2017, attributed to a reduction in smoking-related risk factors [2]. Meanwhile, United States depicted a similar trend which is due to significant advancements in evidence-based therapies, including thrombolysis, CABG, coronary angioplasty and stents, ACE inhibitors, and statins [9] In west countries, the decline in rates of cardiovascular mortality over the last four decades is secondary to several factors such as changes in population risk factors and improved health care [10].

The epidemiological features of the CVD epidemic in Asia further support the data in this study, which is continuous increase in CVD mortality rate, with a huge proportion of premature CVD deaths. The responsible CVD risk factors in most Asian countries are inadequate management of unhealthy diet, smoking, obesity, hypertension, dyslipidemia, and diabetes, along with the rapid progression of population aging in many Asian countries [13] Specifically in Southeast Asia, age-standardized CVD mortality rates range from 124.9 to 421.6 per 100,000 in 2021 and high systolic blood pressure accounted for the largest proportion of DALYs among all CVD risks, contributing to 56.9% death [14].

## Identifying key factors contributing to escalating mortality rates

This study found that, in addition to an increase in cardiovascular disease mortality, there is an increase in the summary exposure value of several major risk factors for the occurrence and death from cardiovascular disease such as hypertension, obesity, diabetes mellitus, dyslipidaemia, and smoking.

## Hypertension

In this study, we found that hypertension had the most significant number of risk variables when compared to other risk factors. Furthermore, hypertension has been dominating the most attributable CVD death among other risk factors for the past 19 years. Data from Basic Health Research (RISKESDAS) demonstrate that the prevalence of hypertension according to physician diagnosis and medication accounted for 8.36% and 8.84% respectively. On contrary, based on the blood pressure measurement, the prevalence of hypertension reaches as high as 34.11% (RISKESDAS). This depicts the occurrence of the condition called underdiagnosis in Indonesia. Other study also confirmed this state and identified that rural areas have a higher rate of underdiagnosis and higher wealth is more likely to undergo regular medical checkups and diagnosed as hypertension [15]. Besides, RISKESDAS also found that a significant portion of hypertension patients in Indonesia, around 32.27%, do not consume medication routinely, with 13.33% not taking any medication at all. The reasons are mainly because many people do not feel any symptomps (59.8%), lack of regular control (31.1%), the use of other traditional medications (14.5%), and forgetfulness (11.5%). Only 12% of all hypertension patients in Indonesia routinely measure their blood pressure (RISKESDAS). Undermedication for hypertension is important issue that also needs to be addressed. Study revealed that lower wealth groups were more likely to be undermedicated, particularly in rural areas [15]. Other risk factors such as older age, lower education levels, overweight or obesity, and healthcare utilization were associated with hypertension high number in Indonesia [16]. In terms of hypertension in adolescents, lifestyle and behavior changes, along with sex, family, nutritional status, physical activity, and most significantly, stress, were associated with hypertension occurrence among Indonesian adolescents [17]. Another study focused on coastal communities in Indonesia revealed increasing age and individuals with lower socioeconomic status had a higher risk of

hypertension [18]. In fact, despite the high prevalence, hypertension is not always identified as the leading cause of death in certain studies, while playing a critical role in the development and progression of CVD [19].

In this study, hypertension is revealed to have a higher prevalence than any other risk factor; yet there does not appear to be a meaningful association between changes in hypertension exposure and rising mortality rates in Indonesia. Furthermore, it has been demonstrated that high LDL exposure in men has a substantial link with an increase in mortality, as does diabetes mellitus in women.

## Obesity

This study highlights that the highest cardiovascular death portion is attributed to obesity or high BMI. High BMI is shown to be responsible to substantial number of deaths, primarily caused by CVD [20]. A review in general population has also depicted that obesity or high BMI is associated with an increased risk of CVD mortality [21]. Obesity plays an important role in CVD mortality by directly contributing to the development of other cardiovascular risk factors such as dyslipidemia, type 2 diabetes, hypertension, and sleep disorders [22]. Moreover, obesity is also evidenced to independently lead to cardiovascular mortality irrespective of other cardiovascular risk factors [23]. Adiposity, especially visceral or abdominal fat, is strongly linked to cardiovascular risk and disease through various physiological mechanisms, including metabolic, endocrinologic, immunologic, structural, humoral, hemodynamic, and functional alterations [22]. Furthermore, obesity is strongly associated with DKOLH-CVD (diabetes, chronic kidney disease, obesity, hyperlipidemia, or hypertensive heart disease). Regarding other countries data, DKOLH-CVD exhibits a significant proportion of CVD mortality in the US and Australia. Recent data indicate that the mortality rates associated with DKOLH-CVD have been increasing, especially at younger ages [24].

Overall, global trends in CVD incidence and mortality rates indicate a decline in most countries since 1990. However, developed nations have experienced a more significant reduction, while developing nations still face challenges in reducing the burden [25]. By examining the global and regional patterns of cardiovascular disease, it becomes evident that various factors such as economic transitions, lifestyle changes, and access to healthcare contribute to the varying trends observed across different countries and regions. These insights are crucial for the development of targeted prevention strategies and further research to address the challenges faced by different nations in combatting cardiovascular disease [1, 2, 9, 11, 12].

These findings could be attributed to the high-risk factor for Indonesian adults aged >40 years and the rates of preventive therapy are low [26]. In the Southeast Asian Region (SEAR), it has been observed that the onset of cardiovascular disease (CVD) tends to occur at a relatively young age. This means that individuals in this region are experiencing the development of CVD earlier in their lives compared to other regions [27]. Additionally, there is a concerning trend of premature death resulting from CVD, which is growing at a rapid pace. This alarming increase in premature CVD-related deaths further emphasizes the urgent need for effective preventive measures and interventions in the SEAR [26]. There is a notable recognition in high body mass index or obesity which is showed to dramatically increasing from 9% to 18% between 2000 to 2019 which is accounted for 37,864 and 118,188. Obesity's SEV rate dramatically soared up from 6,7 to 16 (+9,3%). The results obtained from the comprehensive analysis conducted in this study have revealed a rather disconcerting pattern in the mortality rates associated with cardiovascular disease. Furthermore, it has been observed that this distressing trend is closely correlated with a notable surge in the overall level of exposure to various significant risk factors that are known to contribute the development and progression of

this condition. The study sheds light on several key factors that play a crucial role in the development and mortality rates associated with cardiovascular disease. These factors include hypertension, obesity, diabetes mellitus, dyslipidaemia, and smoking. Each of these contributors has been found to have a substantial impact on the occurrence and severity of cardiovascular disease, ultimately leading to increased rates of mortality. We uncover a significant correlation between the escalation of plasma glucose exposure in women and a progressively higher deaths caused by ischemic heart disease. On the other hand, it is important to note that when there is an increase in LDL exposure, there is a notable and significant correlation that emerges. The data clearly indicates that as LDL exposure rises, there is a corresponding escalation in mortality rates from this cardiovascular condition.

## Hyperglycemia

The present study demonstrates a strong association between elevated plasma glucose levels and increased death rates attributed to IHD among female individuals. This discovery aligns with the findings of a prior investigation conducted in Brazil, which demonstrated a positive association between elevated plasma glucose levels and cardiovascular mortality among women [2]. The higher prevalence of impaired glucose tolerance in women compared to males, as observed in a previous study conducted in Indonesia by the National Basic Health, is attributed mainly to Diabetes Mellitus [28]. The prevalence of Diabetes Mellitus has been observed to be positively associated with the incidence of obesity. This relationship is particularly evident in the period spanning from 2000 to 2019, during which the prevalence of obesity has demonstrated a consistent upward trend. Additionally, it has been noted in several studies that the prevalence of undiagnosed diabetes mellitus exceeds that of diagnosed cases. We also performed a sensitivity analysis to examine the relationship between elevated plasma glucose levels and the presence of Type II Diabetes Mellitus. The results of the analysis indicate a statistically significant association between these two variables and the increase in IHD mortality. There is evidence to suggest a positive association between those diagnosed with diabetes mellitus and increased rates of both all-cause death and cardiovascular mortality [6]. High fasting plasma glucose is defined by the Institute for Health Metrics and Evaluation (IHME) as exceeding the theoretical minimum risk exposure limit (TMREL), specifically falling within the range of 4.8–5.4 mmol/L or 86.4–97.2 mg/dL. However, another study established a correlation between hbA1C levels and cardiovascular disease (CVD) mortality, specifically when the hbA1C value exceeds or is equal to 7 [29].

## Dyslipidemia

Similar trends have been observed in other developing nations across the globe, including the bustling city of Beijing and the beautiful archipelago of the Philippines. These nations, like many others, have experienced significant increases in total cholesterol levels, which is a key indicator of cardiovascular health. Additionally, there has been a higher prevalence of atherosclerosis risk factors in these regions, highlighting the urgent need for preventive measures and interventions to address this growing health concern.

This study found that, in addition to an increase in cardiovascular disease mortality, there is an increase in the summary exposure value of several major risk factors for the occurrence and death from cardiovascular disease, such as hypertension, obesity, diabetes mellitus, dyslipidemia, and smoking. This rise was observed in both genders, namely women and men. In this study, we found that hypertension had the most significant number of risk variables when compared to other risk factors. Nonetheless, the findings in this paper suggest that there is a significant association between the change in high plasma glucose exposure in women and the

increasing number of ischemic heart disease deaths. We discovered a strong link between increased LDL exposure and rising ischemic heart disease death rates in males. These findings can be explained by the fact that, despite the high prevalence of hypertension, hypertension is not always identified as the leading cause of death in certain studies, while playing a critical role in the development and progression of CVD [19]. These findings are also consistent with previous studies in which men were more likely to smoke, eat a high-risk diet, and have a low HDL cholesterol level, whereas women were more likely to have a low physical activity level, the presence of mental-emotional disorders, obesity, a high WC, a high WtHR, hypertension, diabetes, a high total cholesterol level, and a high LDL cholesterol level [6].

Another research in a developing nation, Beijing, found the same thing, with the majority of the increase ascribed to significant increases in total cholesterol levels [12]. Additionally, it is comparable in other developing nations, such as the Philippines, where the prevalence of atherosclerosis risk factors was higher in 2018 than in 2003, despite a non-significant increase in diabetes and a drop in smoking [30].

## Understanding the factors behind mortality decline in developed nations

The results here contrast with those of some affluent nations, such as the United States and Brazil, where fatalities from cardiovascular disease are falling [2, 3]. According to research done in Brazil, there has been a decline in cardiovascular disease mortality from 2005 to 2017, followed by a trend of reducing risk factors for smoking [2]. Meanwhile, in America, significant advances in evidence-based therapies, such as the use of thrombolysis, coronary artery bypass grafting (CABG), coronary angioplasty and stents, angiotensin-converting-enzyme (ACE) inhibitors, statins, have revolutionized the treatment of cardiovascular disease [3].

Another comprehensive research study was undertaken to delve deeper into the factors contributing to the decline in cardiovascular mortality rates in Maryland, USA. The significant decrease in cardiovascular disease rates can be attributed to the remarkable advancements in prevention and treatment strategies. These advancements have led to a rapid decline in cigarette smoking, better management of hypertension, widespread utilization of statins to reduce cholesterol levels, and the timely implementation of thrombolysis and stents in cases of acute coronary syndrome to minimize or even prevent heart attacks [31]. The prevention and control of main CHD risk factors, successful treatment of existing CHD, and possibly other unidentified factors affect CHD death rates. Due to the magnitude of these factors and their success in preventing CHD deaths, CHD trends vary by age, sex, race, ethnicity, geographic location, and other socio-demographic categories. Thus, CHD death rates may continue to drop in some populations at the spectacular rates seen in the US in the early 1970s [3].

In this study, we also investigate by incorporating many variables that potentially affect these risk factors into the multivariate model, including the covariates Socio-demographic Index, Primary health care coverage, and GDP per capita. Women still received the same results after we added these variables; that is, there was a substantial effect on mortality from cardiovascular disease and exposure to diabetes risk factors. After adjusting for this covariate in males, the research findings demonstrate a substantial effect of hypertension and obesity on the rise in stroke mortality in Indonesia.

We also continue the year lag analysis to see whether there is a relationship between risk variables in prior years and the rise in cardiovascular events in Indonesia. The findings of this study revealed that the risk factors for dyslipidaemia in the previous 2,5,8,10 years had a substantial effect on the rise in mortality from ischemic heart disease in both men and women. These findings differ from research in Brazil, in which practically all risk variables in the year-lag analysis contributed to a decrease in CVD mortality [2].

We hereby acknowledge some study limitations. This analysis included observational studies by independent researchers and nationwide government survey to assess risk factor exposure. Thus, those exposed to risk factors may have had a pre-existing cardiovascular disease, which could affect our findings. Despite the need to account for exposure and outcome measurement error, the Institute for Health Metrics and Evaluation (IHME) and Global Burden of Disease (GBD) studies have the largest databases in their disciplines. These findings support a large prospective study that linked modifiable risk variables to cardiovascular disease (CVD) mortality in 21 countries. In this study, multiple covariates were incorporated, including SDI, GDP per capita, and primary health care coverage. The data was categorized based on provinces and sourced from government records. However, it is important to note that in Indonesia, there was a modification in the number of provinces. Specifically, in 2012, two additional provinces, East Kalimantan, and North Kalimantan, were established, resulting in a total of 34 provinces. Consequently, there is a possibility of shared data between these two provinces, which could potentially influence the outcomes of this study.

The application of this research implies that the government should have become more aware of metabolic and behavioural risk factors as early as ten years ago and enhanced the regulation of preventive approaches and initiatives. The government can take proactive actions to lessen the burden of cardiovascular disease by detecting and addressing risk factors such as dyslipidaemia, diabetes mellitus, hypertension, poor diets, physical inactivity, and smoking. This strategy could include implementing comprehensive public health campaigns to promote healthier lifestyles, improving access to healthcare services for early detection and management of risk factors, and strengthening food labelling, tobacco control, and physical activity promotion regulations and policies. The government can successfully address the rising incidence of cardiovascular disease and enhance the overall cardiovascular health of the people by prioritizing preventative measures and taking a multi-sectoral strategy.

This research will help to strengthen Asia's demographic database. It will provide useful insights into the health profiles of Asian communities by evaluating the prevalence and impact of cardiovascular risk factors such as dyslipidaemia, diabetes mellitus, hypertension, and behavioural variables. The findings can assist inform public health policy, resource allocation, and tailored interventions to address the Asian population's special cardiovascular health requirements. Furthermore, by expanding the knowledge base on cardiovascular risk factors across Asia, this research will help to improve global understanding of cardiovascular disease and support efforts to develop more effective prevention and management strategies globally.

## Conclusion

Hyperglycemia in women is an important risk factor associated with increasing mortality due to Ischemic Heart Disease (IHD) in Indonesia. A year lag analysis, on the other hand, revealed that high LDL levels in both men and women were associated with an increase incidence of IHD. Additionally, the highest cardiovascular death portion were attributed to obesity. These findings suggest that policymakers should prevent high LDL risk factors and hyperglycemia 10 years earlier and employ personalized therapy to regulate associated risks. The information gathered of this study will help to supplement the Asian population database.

## Supporting information

**S1 File.**
(DOCX)

## Acknowledgments

Authors are thankful to the research colleague in the Advanced Research Center Team at the Faculty of Medicine, Airlangga University, Indonesia

## Author Contributions

**Conceptualization:** Wigaviola Socha Purnamaasri Harmadha, Zahras Azimuth.

**Data curation:** Wigaviola Socha Purnamaasri Harmadha, Zahras Azimuth, Hanif Ardiansyah Sulistya, Fikri Firmansyah.

**Formal analysis:** Renato Simoes Gaspar, Fikri Firmansyah.

**Investigation:** Farizal Rizky Muharram.

**Methodology:** Wigaviola Socha Purnamaasri Harmadha, Renato Simoes Gaspar.

**Project administration:** Wigaviola Socha Purnamaasri Harmadha.

**Resources:** Wigaviola Socha Purnamaasri Harmadha, Hanif Ardiansyah Sulistya, Fikri Firmansyah.

**Software:** Wigaviola Socha Purnamaasri Harmadha.

**Supervision:** Farizal Rizky Muharram, Renato Simoes Gaspar, Rendra Mahardika Putra.

**Visualization:** Wigaviola Socha Purnamaasri Harmadha.

**Writing – original draft:** Wigaviola Socha Purnamaasri Harmadha.

**Writing – review & editing:** Wigaviola Socha Purnamaasri Harmadha, Zahras Azimuth, Hanif Ardiansyah Sulistya, Chaq El Chaq Zamzam Multazam, Muhammad Harits, Rendra Mahardika Putra.

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
