## [Decision Letter · Decision Letter 0]

24 Aug 2023

PONE-D-23-23346Explaining the Increase Mortality from Cardiovascular Disease in Indonesia: Unraveling the trends in the Influence of Metabolic and Behavioral Risk Factors (2000-2019)PLOS ONE

Dear Dr. Mahardhika,

Thank you for submitting your manuscript to PLOS ONE. After careful consideration, we feel that it has merit but does not fully meet PLOS ONE’s publication criteria as it currently stands. Therefore, we invite you to submit a revised version of the manuscript that addresses the points raised during the review process.

We look forward to receiving your revised manuscript.

Kind regards,

Seyed Aria Nejadghaderi

Academic Editor

PLOS ONE

Journal Requirements:

4. Please ensure that you refer to Figure 3 in your text as, if accepted, production will need this reference to link the reader to the figure.

5. Please remove your figures from within your manuscript file, leaving only the individual TIFF/EPS image files, uploaded separately. These will be automatically included in the reviewers’ PDF.

6. Please include your tables as part of your main manuscript and remove the individual files. Please note that supplementary tables (should remain/ be uploaded) as separate "supporting information" files

Reviewers' comments:

Reviewer's Responses to Questions

**Comments to the Author**

1. Is the manuscript technically sound, and do the data support the conclusions?

Reviewer #1: Yes

Reviewer #2: Yes

Reviewer #3: No

2. Has the statistical analysis been performed appropriately and rigorously? 

Reviewer #1: Yes

Reviewer #2: Yes

Reviewer #3: No

3. Have the authors made all data underlying the findings in their manuscript fully available?

Reviewer #1: Yes

Reviewer #2: Yes

Reviewer #3: No

4. Is the manuscript presented in an intelligible fashion and written in standard English?

Reviewer #1: Yes

Reviewer #2: Yes

Reviewer #3: No

5. Review Comments to the Author

Reviewer #1: Authors used data from IHME and evaluated trends in cardiovascular mortality in Indonesia (from 2000 to 2019). The study is well-designed and can provide helpful information for policymakers and can enlighten further research. Although the study is well-written, some points need to be addressed:

1- Abbreviations should be defined in their first use.

2- Limitations of the study have not been mentioned. In the last paragraph of the discussion, mention all limitations of this study. For example, data is from one database (IHME) and limits its generalizability. Moreover, no subnational analysis was performed in this study.

3- The authors used and analyzed data from GBD IHME. They should clearly mention this in the title, abstract, and main text. Please revise comprehensively.

4- First paragraph of the discussion section should mention main findings of this study.

5- Define abbreviation in figures and tables in their captions.

Reviewer #2: In this study by Socha et al., the authors analyzed the incidence rate, deaths, and DALYs of cardiovascular diseases (CVDs) from 2000 to 2019. They also examined the association between various prevalent behavioral and metabolic risk factors (hypertension, hyperglycemia, obesity, dyslipidemia, and smoking) and CVDs, categorizing their results by gender.

Although this study is interesting, there are several concerns, and I have a few critical questions and comments:

The conclusion and methodology of the abstract are confusing and need revision to improve clarity.

In the first paragraph of the introduction, the authors present paradoxical sentences. Please clarify whether you are discussing the prevalence of CVDs or the rate of change in the incidence of CVDs among developing and developed countries.

There are several errors in the reference numbers. Please correct the references. Additionally, some sentences lack references, while others with references seem to be incorrect. Therefore, it is necessary to revise the references throughout the manuscript.

The last paragraph of the introduction does not align with the stated aim in the abstract. Please revise these sections. Are you discussing an increase or decline in the incidence of CVDs?

Use the complete form of abbreviations in their first usage and then only use the abbreviations afterwards. Avoid repeating the complete form throughout the text.

In Figure 1, between 2015 and 2019, the mortality rate and DALYs for CVDs showed a decrease, especially in mortality. Please mention this observation in your results and discussion.

It would be valuable if you could report and compare these results based on age groups in addition to gender.

The results were not adequately discussed. Please provide one or two paragraphs regarding the metabolic and behavioral changes during recent years and their impact on CVDs. Discuss your results based on previous knowledge and elaborate on the differences between males and females based on your findings. Furthermore, provide several studies similar to yours, discuss their results, and compare them to your findings.

Please explain the limitations of your study.

Finally, provide a summary of your conclusion and future prospects at the end of the discussion section.

There are several typographical, grammatical, and English usage errors throughout the manuscript. Please revise them.

Reviewer #3: Dear Authors

I would like to express my gratitude for being involved in reviewing this interesting manuscript. After scrutinizing it; there are some points to be considered;

# Title and abstract section

• Authors should indicate the study’s design with a commonly used term in the title or the abstract.

• I am wondering about the sentence written in the abstract: Over the past two decades, CVD burden has shifted from developing to developed countries, this is rather counterintuitive. The authors should reexamine the references.

• The conclusion was rather lengthy, and the authors should focus on the results of the analysis

# Introduction section

• In the # Introduction, the authors did not explore the knowledge gap in terms of the association between behavioral and CVD risk factor with CVD in Indonesia; there is no bridging between # Paragraph 4 and # Paragraph 5.

• I am wondering the way the authors write the references quoted, the authors should jot down in order.

# Cardiovascular Diseases Outcome section:

• The authors stated that mortality rate, DALY to CVD, ischemic heart disease, stroke by gender, province and year were considered as independent variables; I would suggest that those are not suitable with the objective of the study. The authors should consider it.

# Multivariate regression

• In #Multivariate regression section, the authors stated that CVD as an outcome variable, this contradicts with the previous statement

• The authors would be much better in compiling the manuscript following the IMRAD, and the #multivariate regression should be put under the heading of #Statistical analysis

# Conclusion, I am wondering the authors did not elaborate the conclusion of the study and the limitation.

6. PLOS authors have the option to publish the peer review history of their article (what does this mean?). If published, this will include your full peer review and any attached files.

Reviewer #1: No

Reviewer #2: **Yes: **Malihe Rezaee

Reviewer #3: No

---

## [Author Response · Author response to Decision Letter 0]

2 Oct 2023

Reviewer #1: 

Authors used data from IHME and evaluated trends in cardiovascular mortality in Indonesia (from 2000 to 2019). The study is well-designed and can provide helpful information for policymakers and can enlighten further research. Although the study is well-written, some points need to be addressed:

1. Abbreviations should be defined in their first use. 

• Respond: I have added abbreviation required in the first use.

2. Limitations of the study have not been mentioned. In the last paragraph of the discussion, mention all limitations of this study. For example, data is from one database (IHME) and limits its generalizability. Moreover, no subnational analysis was performed in this study.

• Respond: I have added limitations of the study in the last paragraph of the discussion.

3. The authors used and analyzed data from GBD IHME. They should clearly mention this in the title, abstract, and main text. Please revise comprehensively.

• Respond: I have added Global Burden Study in the title. The title is now “Explaining the Increase of Incidence and Mortality from Cardiovascular Disease in Indonesia: A Global Burden of Disease Study Analysis (2000-2019)”.

4. First paragraph of the discussion section should mention main findings of this study.

• Respond: I have mentioned main findings and break down each finding in the discussion section.

5. Define abbreviation in figures and tables in their captions.

• Respond: I have added abbreviation definition below figures and tables. 

Reviewer #2: 

In this study by Socha et al., the authors analyzed the incidence rate, deaths, and DALYs of cardiovascular diseases (CVDs) from 2000 to 2019. They also examined the association between various prevalent behavioral and metabolic risk factors (hypertension, hyperglycemia, obesity, dyslipidemia, and smoking) and CVDs, categorizing their results by gender.

Although this study is interesting, there are several concerns, and I have a few critical questions and comments:

1. The conclusion and methodology of the abstract are confusing and need revision to improve clarity.

• Respond: I have revised the conclusion and methodology and improve the clarity of the abstract. The sentence is probably confusing, but what I am trying to address in year lag analisis of the conclusion is that this study also revealed that the presence of high LDL levels in both men and women were associated with an increase incidence of IHD that manifested several years subsequent to exposure to those risk factors.

2. In the first paragraph of the introduction, the authors present paradoxical sentences. Please clarify whether you are discussing the prevalence of CVDs or the rate of change in the incidence of CVDs among developing and developed countries.

• Respond: I have clarified in the introduction sentence, what I am trying to employ is that in the developing countries the burden of CVD is starting to increase while in the well-developed countries are the opposite. This indicates that we need to investigate what change could contribute to those changes. 

3. There are several errors in the reference numbers. Please correct the references. Additionally, some sentences lack references, while others with references seem to be incorrect. Therefore, it is necessary to revise the references throughout the manuscript.

• Respond: I have corrected those errors.

4. The last paragraph of the introduction does not align with the stated aim in the abstract. Please revise these sections. Are you discussing an increase or decline in the incidence of CVDs?

• Respond: I have already corrected my mistake in writing. In this paper, I am discussing about the increase of CVD burden in Indonesia.

5. Use the complete form of abbreviations in their first usage and then only use the abbreviations afterwards. Avoid repeating the complete form throughout the text.

• Respond: I have revised this.

6. In Figure 1, between 2015 and 2019, the mortality rate and DALYs for CVDs showed a decrease, especially in mortality. Please mention this observation in your results and discussion.

It would be valuable if you could report and compare these results based on age groups in addition to gender.

• Respond: I have added this to the result. 

7. The results were not adequately discussed. Please provide one or two paragraphs regarding the metabolic and behavioral changes during recent years and their impact on CVDs. Discuss your results based on previous knowledge and elaborate on the differences between males and females based on your findings. Furthermore, provide several studies similar to yours, discuss their results, and compare them to your findings.

Please explain the limitations of your study.

• Respond: I have changed the discussion by making a detail paragraph on each risk factor.

8. Finally, provide a summary of your conclusion and future prospects at the end of the discussion section.

• Respond: I have added this in the last paragraph of the discussion

9. There are several typographical, grammatical, and English usage errors throughout the manuscript. Please revise them.

• Respond: I have changed the errors.

Reviewer #3: Dear Authors

I would like to express my gratitude for being involved in reviewing this interesting manuscript. After scrutinizing it; there are some points to be considered;

# Title and abstract section

• Authors should indicate the study’s design with a commonly used term in the title or the abstract.

• Respond: I have changed the title and revised the abstract to clarify the methodology

• I am wondering about the sentence written in the abstract: Over the past two decades, CVD burden has shifted from developing to developed countries, this is rather counterintuitive. The authors should reexamine the references.

• Respond: I have changed this sentence to avoid confusion.

• The conclusion was rather lengthy, and the authors should focus on the results of the analysis

• Respond: I have shortened the conclusion.

# Introduction section

• In the # Introduction, the authors did not explore the knowledge gap in terms of the association between behavioral and CVD risk factor with CVD in Indonesia; there is no bridging between # Paragraph 4 and # Paragraph 5.

• Respond: I have added this in the introduction in paragraph 4

• I am wondering the way the authors write the references quoted, the authors should jot down in order.

• Respond: I have corrected this

# Cardiovascular Diseases Outcome section:

• The authors stated that mortality rate, DALY to CVD, ischemic heart disease, stroke by gender, province and year were considered as independent variables; I would suggest that those are not suitable with the objective of the study. The authors should consider it.

• Respond: I have corrected this mistake. The risk factors in this study are the independent variables, while the CVD outcomes are the dependent variables.

# Multivariate regression

• In #Multivariate regression section, the authors stated that CVD as an outcome variable, this contradicts with the previous statement

• Respond: I have corrected this mistake same as above.

• The authors would be much better in compiling the manuscript following the IMRAD, and the #multivariate regression should be put under the heading of #Statistical analysis

• Respond: I have changed it to clarify the methodology.

# Conclusion, I am wondering the authors did not elaborate the conclusion of the study and the limitation.

• Respond: I have added this in the discussion in the third paragraph from the end of discussion

---

## [Decision Letter · Decision Letter 1]

25 Oct 2023

Explaining the Increase of Incidence and Mortality from Cardiovascular Disease in Indonesia: A Global Burden of Disease Study Analysis (2000-2019)

PONE-D-23-23346R1

Dear Dr. Mahardhika,

We’re pleased to inform you that your manuscript has been judged scientifically suitable for publication and will be formally accepted for publication once it meets all outstanding technical requirements.

Kind regards,

Seyed Aria Nejadghaderi

Academic Editor

PLOS ONE

Additional Editor Comments (optional):

Reviewers' comments:

Reviewer's Responses to Questions

**Comments to the Author**

1. If the authors have adequately addressed your comments raised in a previous round of review and you feel that this manuscript is now acceptable for publication, you may indicate that here to bypass the “Comments to the Author” section, enter your conflict of interest statement in the “Confidential to Editor” section, and submit your "Accept" recommendation.

Reviewer #1: All comments have been addressed

Reviewer #2: All comments have been addressed

2. Is the manuscript technically sound, and do the data support the conclusions?

Reviewer #1: Yes

Reviewer #2: Yes

3. Has the statistical analysis been performed appropriately and rigorously? 

Reviewer #1: Yes

Reviewer #2: Yes

4. Have the authors made all data underlying the findings in their manuscript fully available?

Reviewer #1: Yes

Reviewer #2: Yes

5. Is the manuscript presented in an intelligible fashion and written in standard English?

Reviewer #1: Yes

Reviewer #2: Yes

6. Review Comments to the Author

Reviewer #1: Thanks for the revision. All comments have been addressed and the paper can be published in its current form.

Reviewer #2: (No Response)

7. PLOS authors have the option to publish the peer review history of their article (what does this mean?). If published, this will include your full peer review and any attached files.

Reviewer #1: No

Reviewer #2: No

---

## [Editor Report · Acceptance letter]

10 Nov 2023

PONE-D-23-23346R1 

Explaining the Increase of Incidence and Mortality from Cardiovascular Disease in Indonesia: A Global Burden of Disease Study Analysis (2000-2019) 

Dear Dr. Putra:

I'm pleased to inform you that your manuscript has been deemed suitable for publication in PLOS ONE. Congratulations! Your manuscript is now with our production department. 

Kind regards, 

on behalf of

Dr. Seyed Aria Nejadghaderi 

Academic Editor

PLOS ONE